# Theoretical Investigation of Interconversion Pathways and Intermediates in Hydride/Silyl Exchange of Niobocene Hydride–Silyl Complexes: A DFT Study Incorporating Conformational Search and Interaction Region Indicator (IRI) Analysis

**DOI:** 10.3390/molecules29215075

**Published:** 2024-10-26

**Authors:** Dapeng Zhang, Naoki Kishimoto

**Affiliations:** Department of Chemistry, Graduate School of Science, Tohoku University, 6-3, Aoba, Aramaki, Aoba-ku, Sendai 980-8578, Japan; zhang.dapeng.c5@tohoku.ac.jp

**Keywords:** niobocene hydride–silyl complexes, hydride/silyl exchange, conformational search, DFT, inter-ligand interaction, interaction region indicator (IRI) analysis

## Abstract

Niobocene hydride–silyl complexes exhibit intriguing structural characteristics with the potential for direct hydride/silyl exchange, where hydride migration plays a crucial role during conformational interconversion. In this study, quantum chemical calculations were utilized to investigate the transformation pathways involved in hydride/silyl exchange in niobocene trihydride complexes with various dichlorosilanes, including SiCl_2_Me_2_, SiCl_2_*^i^*Pr_2_, and SiCl_2_MePh ligands. The conformational changes and hydride shifts within these niobocene hydride–silyl complexes were examined, and key intermediates were identified. Electronic wavefunction analysis provided insights into the coordination configurations and the nature of inter-ligand interactions. Interaction region indicator (IRI) analysis revealed Van der Waals interactions between chloride atoms and cyclopentadienyl rings, as well as between chloride atoms and Me, *^i^*Pr, and Ph groups. Notably, distinct interactions between hydride ligands, including those from Si-H moieties and coordinated hydrogen atoms, were observed. Both lateral and central conformations, with respect to silicon coordination to the niobium center, were considered. This study enhances the understanding of intermediate conformations in the hydride/silyl exchange process and provides a detailed characterization of inter-ligand interactions, offering valuable insights for analyzing metallocene complexes with organic ligand coordination.

## 1. Introduction

Metallocene compounds are defined by their distinctive sandwich structure, wherein a metal atom is coordinated between two parallel cyclopentadienyl (η^5^-C_5_H_5_, abbreviated Cp) rings [1]. This structural configuration, coupled with the unique electronic properties arising from the cation-Cp interaction, confers upon metallocene remarkable catalytic capabilities, particularly in the realm of polymerization reactions [2,3,4,5,6,7,8]. The sandwich configuration of metallocene can undergo distortion and bending upon the formation of hydride derivatives, resulting in metal-hydride species that serve as important intermediates in various reactions. These structural features, induced by hydride coordination, along with the electronic properties of the M-H bonds, are key factors governing the activity and selectivity of the resulting complexes [9,10,11,12,13,14,15,16,17,18]. Among these compounds, niobocene hydride derivatives are noteworthy for their distinctive properties, especially in insertion reactions. When the Nb-H bond of Nb(η^5^-C_5_H_4_SiMe_3_)_2_(H)(P(OMe)_3_) interacts with activated alkynes such as RC≡CR (R = CO_2_Me, CO_2_^t^Bu), a variety of product complexes can form through distinct intermediates [19]. The insertion mechanism involving niobocene hydride complex (Cp_2_NbH_3_) is highly dependent on the substituents of alkyne, specifically their electron-withdrawing (MeO_2_CC≡CCO_2_Me) or electron-donating (MeC≡CMe) characteristics [20].

A distinctive structural characteristic is observed in niobocene hydride complexes featuring silyl ligands directly coordinating to the niobium center. These complexes, which conform to the general formular Cp_2_Nb(SiHMe_2_)H(SiXMe_2_) and Cp_2_Nb(SiYMe_2_)H(SiXMe_2_), exhibit both conventional metal-hydride (M-H) bonds and nonclassical inter-ligand Nb-H···Si-X interactions, where X and Y represent halogen atoms (F, Cl, Br, or I, X ≠ Y) [21,22,23]. The complex Cp_2_Nb(SiXMe_2_)(H)(SiYMe_2_) exhibits noteworthy structural distortions and inter-ligand hypervalent interactions (IHI), characterized by competition between halosilyl groups for interaction with the metal-hydride. This intensifies with the substitution of heavier halogen. The concurrent Si···H interactions give rise to these distortions, which significantly influence key structural parameters, including Nb-Si and Si-X bond lengths, as well as YSi-Nb-SiX bond angles [23]. The coordination configuration of hydride and silyl ligands relative to the bisecting plane of niobocene reveals a distinctive inter-ligand interaction. This spatial configuration of ligands around the niobium center gives rise to the central and lateral isomers in monosilyl and bis(silyl) derivatives. These isomers exhibit varying degrees of stability, which may be modulated by electronic factors that mitigate repulsive interactions in multi-ligand coordination processes [24,25]. Cp_2_NbH_3_ exhibits varied reactivity patterns with dichlorosilanes (SiCl_2_RR’) bearing different substituent groups. When R and R’ are Me or Ph groups, the isomers form in equimolar ratios. However, substitution with the sterically demanding isopropyl (*^i^*Pr) group yields a mixture of unidentified products accompanied by dihydrogen evolution [25]. This observation suggests that the substituents on the dichlorosilanes play a crucial role in mediating specific interactions with the niobocene hydride complexes and may influence the coordination process through different intermediates, thereby forming distinct products. Nevertheless, the precise nature of the intramolecular interactions, the potential isomerization pathways between central and lateral isomers, and the role of the hydride during coordination formation remain unclear. Further investigation into the potential intermediates formed during these reactions is warranted to elucidate the underlying mechanisms.

The intricate formation process of niobocene hydrosilane compounds necessitates an in-depth investigation into the stability conferred by multi-ligand arrangements, with particular attention to their interconversion mechanisms between lateral and central configurations, as well as the intermediates involved in the process leading to alternative hydride coordination modes. Additionally, elucidating the electronic effects that influence ligand arrangements and interaction establishment during the formation of intermediates is essential. To address these critical aspects, we conducted density functional theory (DFT) calculations to gain insights into the energetics and electronic structures of potential intermediates in niobocene hydrosilane complex systems. Intermediates were generated by applying artificial forces to drive the reaction of the selected reactants. Upon identification of a stable complex, substituent groups were systematically modified to obtain the initial geometries for the efficient optimization of analog complexes. A conformational search algorithm was employed to identify energetically favorable conformations, including potential intermediates and products resulting from coordination formation. This conformational exploration approach has been successfully applied to organometallic complexes involving metal-histidine and metal-tripeptide interactions [26,27,28]. To the best of our knowledge, this study represents the first application of this method to a niobocene hydrosilane complex. This study presents a systematic investigation of the conformational interconversion pathways among potential intermediates and analyzes the nature of inter-ligand interactions in niobocene hydride–silyl complexes. Through detailed computational analysis, we elucidate the mechanistic details of complex formation and establish a robust theoretical framework for understanding the fundamental factors that govern conformational stability and reactivity in these multi-ligand systems. Our findings not only advance the understanding of niobocene hydride–silyl complexes but also provide valuable insights into the rational design of related organometallic systems.

## 2. Computational Details

The artificial force induced reaction (AFIR) method [29], implemented in the global reaction route mapping (GRRM, version 17) program [30,31], was employed to facilitate reaction progression and generate structures of intermediates and products. Artificial forces can facilitate molecular fragment combination by effectively lowering the activation barrier between reactant pairs on the diatomic potential energy surface. In each model system, forces of approximately 60–100 kJ/mol were applied to niobium and silicon atoms to overcome the reaction energy barrier. This approach eliminates the need for manual construction of model complexes. AFIR calculations were performed using the TPSSh hybrid *meta*-GGA functionals [32,33] in conjunction with the def2-SVP basis set [34,35] (TPSSh/def2-SVP). Subsequently, the structures were further refined using the TPSSh functional, augmented with Grimme’s dispersion correction (D3) and the Becke–Johnson damping [36,37], along with the def2-TZVPP basis set (TPSSh-D3(BJ)/def2-TZVPP) [34,35]. The TPSSh functional, augmented with D3(BJ) correction demonstrates robust performance for transition metal complexes and provides reliable results for kinetic analyses and mechanistic investigations [38,39,40]. The implementation of the high-accuracy def2-TZVPP basis set enables accurate representation of the electronic structure of metal complexes [41,42]. Based on the optimized structures, constrained conformational explorations were performed using the anharmonic downward distortion following the (ADDF) algorithm [43,44,45,46] implemented in the GRRM program. The ADDF calculation employs a systematic approach to identify anharmonic downward distortions around a given minima on potential energy surface (PES). Larger downward distortions typically indicate pathways to additional stable minima or dissociation channels. Once an initial stable conformation is determined, the identification of other conformers becomes more efficient, as the exploration follows a defined number of the largest downward distortions on the PES. Pathways with smaller downward distortions can be excluded from consideration, as they are unlikely to connect to other stable minima. To enhance computational efficiency in generating equilibrium structures (EQs), transition state structures (TSs) were omitted during the calculation by employing the *EQOnly* option. For each stable structure identified, the ten largest anharmonic downward distortions (*l-ADD10*) were systematically traced to locate surrounding stable minima. The initial conformational search was conducted at the TPSSh/def2-SVP level of theory. Subsequently, the identified EQs were subjected to further refinement at the more robust TPSSh-D3(BJ)/def2-TZVPP level of theory. The interconversion pathways were determined through a combined application of the sphere-contracting walk (SCW) method [47] and the two-point scaled hypersphere search (2PSHS) approach [48]. The SCW method facilitated the identification of intermediates between EQs, while the 2PSHS approach was employed to locate the TSs connecting two EQs. Both EQ and TS structures were calculated at the TPSSh-D3(BJ)/def2-TZVPP level of theory. All structural optimizations [49], frequency analyses, conformational explorations, and calculations related to reaction pathways were conducted using the GRRM program interfaced with Gaussian 16 [50], at a temperature of 298.15 K. The Multiwfn program [51] was used to perform the electronic wavefunction analysis of the calculated complexes. All structural coordinates of the complexes are provided in the Appendix A.

## 3. Results and Discussion

### 3.1. Niobocene Tridydride with Dichlorosilanes SiCl_2_Me_2_, SiCl_2_^i^Pr_2_, and SiCl_2_MePh

The reaction between niobocene trihydride [Cp_2_NbH_3_] (**1**) and chlorosilanes exhibits mono- and bis-substitution processes via hydride/silyl exchange mechanisms, resulting in a range of product complexes featuring either one silyl and two hydride ligands or one hydride and two silyl ligands coordinated to the niobium center [52]. The dichlorosilane SiCl_2_Me_2_ (**2**) reacts with complex **1** via mono(silyl) intermediates (**3**–**5**) to ultimately form complexes (**6**–**8**), as shown in Figure 1. Details of the structural parameters can be found in the Appendix A. Complexes **3**–**5** display similar Nb-H-Si interactions, with angles of 164.24–164.44° and Nb-H/H-Si distances of 1.87–1.89/1.59–1.60 Å. The remaining hydrides coordinate to Nb at 1.72–1.74 Å. Two chlorides bind to Si (2.22–2.32 Å) in each complex, with Nb-Cl distances ranging from 3.63–4.09 Å. Complexes **6**–**8** exhibit direct Nb-Si coordination (2.50–2.53 Å) following chloride and hydride elimination, with preserved Cl-Si bonds (2.19–2.27 Å) and Nb-Si-Cl angles (115.79–116.85°). The remaining hydrogen atoms coordinate to Nb at slightly elongated distances (1.77–1.83 Å) compared to their precursor complexes **3**–**5**. A comparative analysis of the electronic structures reveals that complex **4** exhibits a notably larger HOMO-LUMO gap (4.35 eV) relative to complexes **3** and **5** (see Appendix A). Among these products, complexes **6** and **8** are lateral isomers, while complex **7** represents the central isomer. Complex **7** exhibits distinctive spin-dependent electronic behavior compared to complexes **6** and **8**, with HOMO-LUMO energy gaps of 2.30 eV for α orbitals and 3.51 eV for β orbitals. Notably, in our model complexes, the lateral isomers are deviated from mirror symmetry concerning the Nb-central-hydride plane, resulting in distinct energy differences between these configurations. Specifically, isomers **6** and **8** exhibit an energy difference of 21.7 kJ/mol, attributed to the distinct spatial arrangements of the chloride and two Me groups around the silicon center. The relative energies also suggest that the exchange pathway leading to complexes **6** and **7** is energetically comparable, with values of 36.6 and 36.5 kJ/mol, respectively. These findings align with experimental results [52], which demonstrate that the complexes **6** and **7** can form in an equimolar mixture. In contrast, the lateral-type complex **8** exhibits a higher relative energy of 58.3 kJ/mol, indicating a lower likelihood of formation. However, its intermediate **5** shows a comparable energy to that of the lateral intermediate **3**, at 94.4 and 94.1 kJ/mol, respectively. The central-type intermediate **4** displays a relatively lower energy of 88.9 kJ/mol, suggesting that energy differences are influenced by the positions of chlorosilane ligands relative to the niobium center.

The reaction of complex **1** reacts with SiCl_2_*^i^*Pr_2_ (**9**) proceeds through mono(silyl) intermediates (**10**–**12**) to ultimately form complexes (**13**–**15**), as illustrated in Figure 2. The structural parameters are available in the Appendix A. Complexes **10**–**12** feature Nb-H-Si interactions (159.23–177.22°, near-linear for the central configuration) with Nb-H and H-Si bond distances of 1.88–1.90 Å and 1.60–1.63 Å, respectively. Additional hydride ligands coordinate to Nb at 1.72–1.75 Å. The Si centers bear two chloride substituents (Cl-Si: 2.23–2.31 Å; Nb-Cl: 3.69–3.96 Å). Complex **11** exhibits a larger HOMO-LUMO gap (4.21 eV) relative to complexes **10** and **12** (see Appendix A). In contrast, complexes **13**–**15** exhibit direct Nb-Si coordination (2.50–2.54 Å) with Cl-Si bonds of 2.21–2.28 Å and Nb-Si-Cl angles of 114.92–116.23°. These complexes show elongated Nb-H coordination bonds (1.77–1.84 Å) relative to their respective precursors (complexes **10**–**12**). Complexes **13** and **15** exemplify lateral isomers, whereas complex **14** shows the central configuration. Complex **14** demonstrates distinctive spin-dependent electronic behavior compared to its structural analogues **13** and **15**, with HOMO-LUMO energy gaps of 2.36 eV and 3.35 eV for α and β orbitals, respectively, while complex **13** shows a comparable α-orbital HOMO-LUMO gap of 2.30 eV. The relative energies indicate that the exchange pathway leads to intermediates **10** and **12** with a comparable stability, exhibiting values of 106.5 and 108.5 kJ/mol, respectively. In contrast, the lateral-type intermediate **11** demonstrates a lower relative energy of 91.1 kJ/mol, suggesting a higher propensity for the formation of complex **14**, which possesses the lowest relative energy at 34.2 kJ/mol. Complex 1**5** exhibits a comparable higher energy of 60.8 kJ/mol, indicating a relative instability. Complex **13**, with an intermediate energy of 43.0 kJ/mol, is less likely to form when compared to complex **14**, given the higher energy of its precursor intermediate **10,** relative to intermediate **11**. These energy differences are influenced by the positions of chlorosilane ligands relative to the niobium center. The relative spatial orientation of the chloride and *^i^*Pr substituents results in an observed energy difference of 17.8 kJ/mol between lateral conformations **13** and **15**. Notably, the complexes featuring *^i^*Pr groups exhibit more pronounced energy variations compared to their Me group counterparts, which show less diverse energy differences. This finding suggests that the steric bulk of the *^i^*Pr group significantly impacts the energetics and stability of these complexes, potentially due to increased steric interactions and electronic effects associated with the larger alkyl substituents.

The hydride/silyl exchange of complex **1** upon reaction with SiCl_2_MePh (**16**) involves a series of intermediate structures (**17**–**20**), culminating in the formation of complexes (**21**–**24**), as depicted in Figure 3. Due to steric effects arising from Ph groups, central conformation forms preferentially, while lateral types are prone to dissociation. Complexes **17**–**20** share nearly identical geometric parameters (Appendix A). These species display Nb-H-Si angles of 152.85–153.05° with consistent Nb-H and H-Si bond distances of 1.90 Å and 1.60 Å, respectively. Two chloride substituents are present in each complex, exhibiting Cl-Si bonds of 2.26–2.28 Å and corresponding Nb-Cl distances of 3.89–3.90 Å. The remaining hydride ligands coordinate to the Nb center at 1.74 Å. Similarly, complexes **21**–**24** exhibit consistent structural metrics, featuring Nb-Si bonds of 2.49 Å, Cl-Si distances of 2.21 Å, Nb-Si-Cl angles of 116.30–118.91°, and two Nb-H coordination bonds (1.77–1.79 Å). Due to their central coordination, complexes **17**–**20** exhibit HOMO-LUMO gaps of approximately 4.28 eV, while complexes **21**–**24** display HOMO-LUMO gaps of 1.70 eV for α orbitals and 3.40–3.52 eV for β orbitals (see Appendix A). The intermediates and product complexes exhibit distinct conformations characterized by rotational adjustments of the Cl, Ph, and Me moieties within the chlorosilane ligand. Interestingly, these species exhibit remarkably similar relative energies to the initial reactants: 82.5, 82.0, 84.4, and 82.2 kJ/mol, respectively. The final complexes **21**–**24** also display comparable energetic states, with values of 8.6, 11.4, 14.1, and 11.9 kJ/mol, suggesting a degree of thermodynamic equivalence. Compared to results obtained using SiCl_2_Me_2_ and SiCl_2_*^i^*Pr_2_ ligands, the SiCl_2_MePh ligand imparts less steric hindrance to the intermediates and provides a means to control the formation of centrally coordinated ligands.

### 3.2. Exploration of the Intermediates and Interconversion Pathways

The conformational landscapes of complexes **10**, **11**, **12**, and **18** were systematically investigated using the ADDF algorithm. EQ structures were efficiently identified, and interconversion pathways between stable structures were elucidated through a combination of SCW and 2PSHS calculations. Given the flexibility of the SiCl_2_*^i^*Pr_2_ and SiCl_2_MePh ligands, the selection of EQ structures was constrained to those maintaining the integrity of the Cp rings and preserving the coordination of the niobium center to these rings, thus avoiding dissociation events.

The transformation between lateral and central conformations of [Cp_2_NbH_3_]·SiCl_2_*^i^*Pr_2_ was investigated through the computational analysis of compounds **11** and **12**, as illustrated by the red pathway in Figure 4. The highest energy barrier during this process was approximately 32.3 kJ/mol between EQ1 and EQ2, indicating a minimal steric hindrance from the *^i^*Pr group. This relatively low energy barrier facilitates the interconversion between lateral and central conformations. The transformation between compounds **10** and **11** was further examined, revealing a significant energy difference of 62.8 kJ/mol between the EQ0 and EQ1 structures, as illustrated by the red pathway in Figure 5. This notable energy disparity between the two transformation processes can be attributed to the distinct conformations adopted by the *^i^*Pr groups in each compound.

Furthermore, [Cp_2_NbH_3_]·SiCl_2_*^i^*Pr_2_ undergoes hydride shifts with a low energy difference of around 8.9 kJ/mol to form [Cp_2_NbH_2_·H]·SiCl_2_*^i^*Pr_2_, starting from complex **12**, as depicted by the blue pathway in Figure 4. These results suggest a considerable flexibility of the coordinated hydrides in this compound. The intermediates during HCl dissociation of [Cp_2_NbH_3_]·SiCl_2_*^i^*Pr_2_ were calculated using complex **10**, as shown by the blue pathway in Figure 5. The dissociation proceeds through an intermediate species denoted as [Cp_2_NbH_2_·HCl]·SiCl*^i^*Pr_2_. In this conformation, the central hydride group engages in an interaction with the Cl atom, forming a transient compound prior to the HCl dissociation step, during which the HCl continues to interact with a coordinated hydride group. The lateral hydride groups remain coordinated to the niobium center throughout the process. The analysis of the relative energies of the conformations reveals a large energy difference of 143.1 kJ/mol between the EQ1 and EQ2 structures. This significant energy barrier indicates that the HCl dissociation mechanism is kinetically challenging and likely proceeds via a stepwise pathway involving the formation of the [Cp_2_NbH_2_·HCl]·SiCl*^i^*Pr_2_ intermediate species.

The conformational transformation between lateral and central configurations of [Cp_2_NbH_3_]·SiCl_2_MePh was not observed despite conformation explorations being applied. Two distinct hydride shift pathways were identified through the combination of SCW and 2PSHS calculations, both involving significant energy differences during hydride shifting. The first pathway, illustrated in red in Figure 6, involves a hydride shift with an energy difference of 86.4 kJ/mol between EQ1 and EQ2. This transformation proceeds via a [Cp_2_NbH_2_·HCl]·SiClMePh intermediate, initiated from complex **18**. In this process, the lateral hydride group interacts with the chloride atom to form a transient species prior to HCl dissociation, during which the hydrogen atom maintains interactions with a coordinated hydride group. The second pathway exhibits similar properties, as depicted in blue in Figure 6, which features a hydride shift with a comparable energy difference of 86.7 kJ/mol between EQ1 and EQ2. This pathway also proceeds through a [Cp_2_NbH_2_·HCl]·SiClMePh intermediate, where the hydride shifts result in the formation of HCl that interacts with another coordinated hydride atom.

### 3.3. Electronic Wavefunction Analysis of Inter-Ligand Interactions

Interaction region indicator (IRI) analysis [53], conducted using the Multiwfn program, was employed to elucidate the nature of chemical bonds and weak interactions within the complexes simultaneously. The analyzed results were visualized using the VMD software (version 1.9.3) [54]. In complex **3**, a Cl ligand exhibits Van der Waals (vdW) interactions with the Cp rings, while the chlorine on the opposite side interacts with the coordinated central hydrogen ligand through vdW forces (depicted by solid arrows in Figure 7). The lateral hydride ligand forms a Si-H interaction with notable attraction to the niobium center. Concurrently, the two remaining coordinated hydride ligands demonstrate strong attractive interactions with each other (illustrated by dashed arrows). Complex **4** displays vdW interactions between the chlorine ligands and the Cp rings (solid arrows). The central hydride ligand forms a Si-H bond while exhibiting attraction to the Nb center and repulsion towards the lateral hydrogen atoms (dashed arrows). In complex **5**, one chlorine ligand forms vdW interactions with the Cp rings, while another chlorine atom exhibits weaker vdW interactions with the central hydrogen atom (solid arrows). Simultaneously, silicon forms a bond with a lateral hydride ligand, which, in turn, shows attraction to the Nb center (dashed arrows). Complex **6** demonstrates vdW interactions between the chlorine ligand and the Cp rings, as well as between one Me group and the surrounding Cp ring (solid arrows). Notably, silicon exhibits strong attraction to the Nb center, indicative of chemical bond formation (dashed arrows). In complex **7**, vdW interactions are observed between the edges of two Cp rings and between the Me groups, chlorine group, and Cp rings (solid arrows). Silicon displays strong attraction to the Nb center, suggesting the formation of a chemical bond (dashed arrows). Complex **8** exhibits vdW interactions between the methyl groups, chlorine group, and Cp rings, as well as between the -CH edges of two Cp rings (solid arrows). As in complexes **6** and **7**, silicon shows strong attraction to the Nb center, indicative of chemical bond formation (dashed arrows).

In complexes **10**–**15**, various vdW interactions are observed with distinct patterns for each complex. The results of the analysis are presented in Figure 8. Compared to the results with the Me moieties, the presence of *^i^*Pr groups facilitates the vdW interactions due to their extended moieties, while the chloride atoms contribute to these interactions by engaging with both the *^i^*Pr groups and Cp rings. Complex **10** exhibits vdW interactions between *^i^*Pr groups, chlorine atoms and *^i^*Pr groups, and *^i^*Pr groups and Cp rings. A notable feature is the strong vdW interaction between a chlorine atom and Cp rings. The hydrogen atom of Si-H moiety shows attraction to the niobium center, while the remaining lateral and central hydride ligands also exhibit attractive interactions. Complex **11** presents vdW interactions between *^i^*Pr groups, a chlorine atom and *^i^*Pr groups, *^i^*Pr groups and Cp rings, and chlorine atoms and Cp rings. The hydrogen atom of the Si-H moiety exhibits attraction to the Nb center but repulsion from lateral hydride ligands. In complex **12**, vdW interactions are observed between *^i^*Pr groups, chlorine atoms and *^i^*Pr groups, *^i^*Pr groups and Cp rings, and chlorine atoms and Cp rings. The Si-H group’s hydrogen atom is attracted to the niobium center but repelled by the central hydride ligand, which is simultaneously attracted to another lateral hydrogen atom. Complex **13** demonstrates vdW interactions between ^*i*^Pr groups, the chlorine atom and *^i^*Pr groups, *^i^*Pr groups and Cp rings, and the chlorine atom and Cp rings. Notably, the silicon atom demonstrates strong attraction to the niobium center, potentially indicative of a chemical bond. Complexes **14** and **15** share similar interaction patterns, including vdW interactions between *^i^*Pr groups, the chlorine atom and *^i^*Pr groups, *^i^*Pr groups and Cp rings, and the chlorine atom and a Cp ring. Additionally, -CH groups on one side of the Cp rings exhibit vdW interactions. In both complexes, lateral hydrogen atoms show attraction to the silicon center, which strongly interacts with the metal center, suggesting high-intensity and stable interactions.

In complexes **17**–**24**, various intramolecular interactions were observed (Figure 9). For complexes **17**–**20**, Van der Waals (vdW) interactions were identified between the chlorine and Ph groups with the Cp rings. Additionally, Cl ligands formed vdW interactions with the -CH moieties of the Ph groups. The hydrogen atom of the Si-H group exhibited an attractive interaction with the niobium center, while the remaining lateral hydride ligands formed vdW interactions with the chlorine atoms. Complexes **21**–**24** displayed similar interaction patterns, with some variations. The Ph groups consistently formed vdW interactions with the bottom Cp ring. In complex **21**, additional vdW interactions were observed between the Ph group and both the Me group and a Cl atom. Complexes **22**–**24** showed vdW interactions between the Cl groups and the -CH moieties of the Ph groups, as well as between the -CH edges of the Cp rings. The Me groups in these complexes also formed vdW interactions with the -CH moieties of the Cp rings. A notable difference in complexes **21**–**24** was the strong interaction between the silicon atom and the Nb center, replacing the Si-H···Nb interaction observed in complexes **17**–**20**. Lateral hydrogen atoms in all complexes formed various interactions, contributing to the overall molecular structure. In contrast to the more flexible *^i^*Pr groups, Ph groups exhibit constrained mobility and offer reduced vdW interactions with other ligands. Chlorine atoms predominantly mediate interactions with Cp rings and moieties derived from Ph, *^i^*Pr, and Me groups. In certain instances, chlorine atoms may also participate in vdW interactions with the coordinated hydride ligands. When directly coordinated to the metal center, silicon atoms can form strong attractions with the niobium nucleus while maintaining interactions with other coordinated hydride ligands.

## 4. Conclusions

This study employed DFT calculations, incorporating conformational search and IRI analysis, to investigate niobocene hydride–silyl complexes formed by niobocene trihydride complexes with SiCl_2_Me_2_, SiCl_2_*^i^*Pr_2_, and SiCl_2_MePh ligands. Conformational stability analyses reveal that substituent effects on dichlorosilanes significantly influence the relative stabilities of potential intermediates and product complexes. For SiCl_2_Me_2_, lateral and central intermediates exhibit modest energy differences, with the central configuration showing a slightly higher stability. While lateral and central products demonstrate a comparable stability, the stability of lateral conformers is strongly dependent on the rotational orientation of the Cl and Me substituents. In the case of SiCl_2_*^i^*Pr_2_, the central-type intermediate displays enhanced stability, and lateral intermediates maintain comparable stability. The central-type product shows greater stability relative to lateral conformers; however, the energy barriers between intermediates are notably higher compared to systems with Me or Ph substituents. For the SiCl_2_MePh ligand, the central configuration is exclusively favored, with minor energy variations arising from substituent rotations. While this system exhibits the lowest energy barriers for intermediate interconversion compared to Me- or *^i^*Pr-substituted analogs, the products maintain energy levels comparable to the reactants. The potential transformation processes were systematically investigated. Our findings elucidate the interconversion pathways of stable conformations, revealing potential hydride migration processes and intermediates associated with HCl dissociation. IRI analysis uncovered significant inter-ligand interactions, including Van der Waals interactions among chloride atoms, Cp rings, hydride ligands, as well as Me, *^i^*Pr, and Ph moieties. Notably, in trihydride complexes, silicon atoms formed weaker interactions with the Nb center, while in dihydride complexes, they exhibited strong attraction, forming stable coordination while simultaneously interacting with the remaining hydride ligand. This study advances our understanding of conformational information of intermediates of hydride/silyl exchange, potential transformation pathways related to hydride migrations, and inter-ligand interactions in niobocene complexes, thereby contributing to the broader understanding of organic ligand coordination in metallocene complexes and providing a foundation for future research in this area.

## Figures and Tables

**Figure 1 molecules-29-05075-f001:**
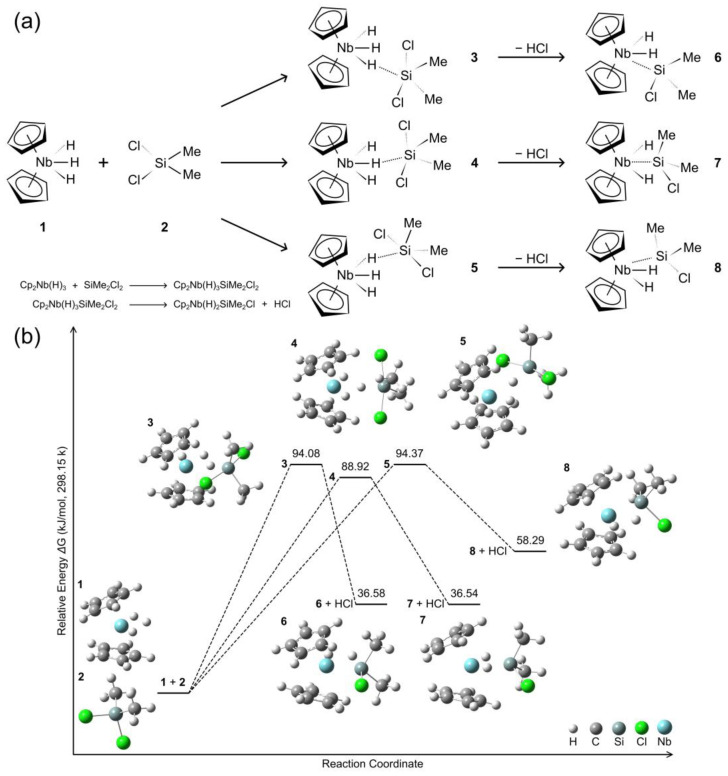
(**a**) Schematic of hydride/silyl exchange process; (**b**) Calculated energy profile for the reaction of [Cp_2_NbH_3_] with Cl_2_SiMe_2_.

**Figure 2 molecules-29-05075-f002:**
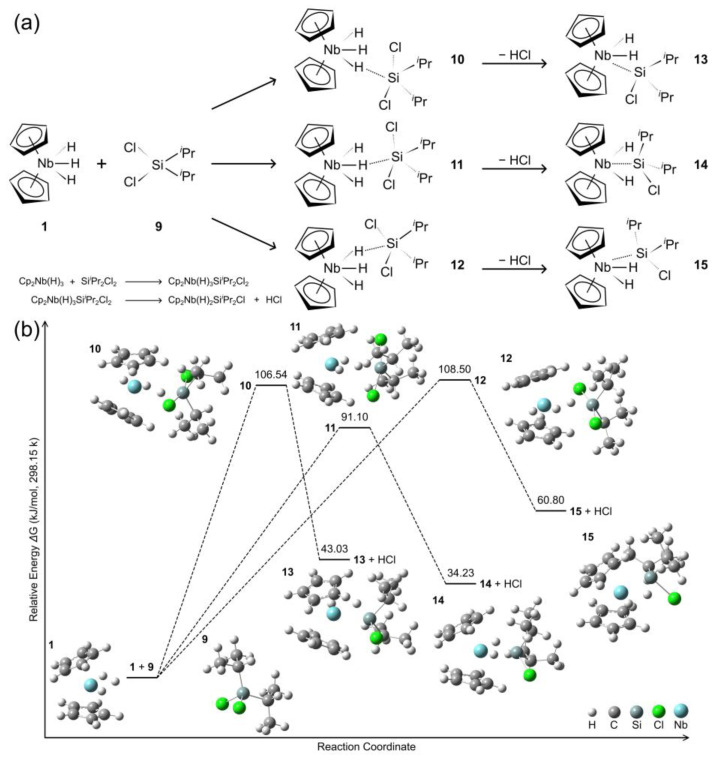
(**a**) Schematic of hydride/silyl exchange process; (**b**) Calculated energy profile for the reaction of [Cp_2_NbH_3_] with Cl_2_Si*^i^*Pr_2_.

**Figure 3 molecules-29-05075-f003:**
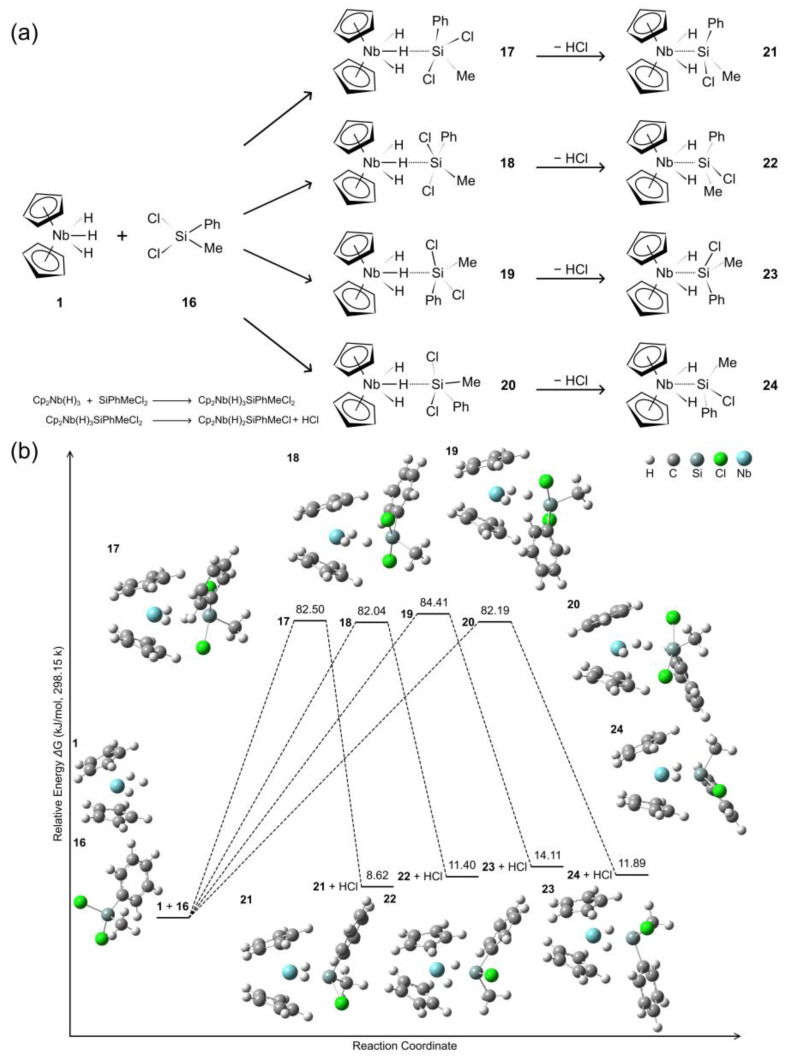
(**a**) Schematic of hydride/silyl exchange process; (**b**) Calculated energy profile for the reaction of [Cp_2_NbH_3_] with Cl_2_SiPhMe.

**Figure 4 molecules-29-05075-f004:**
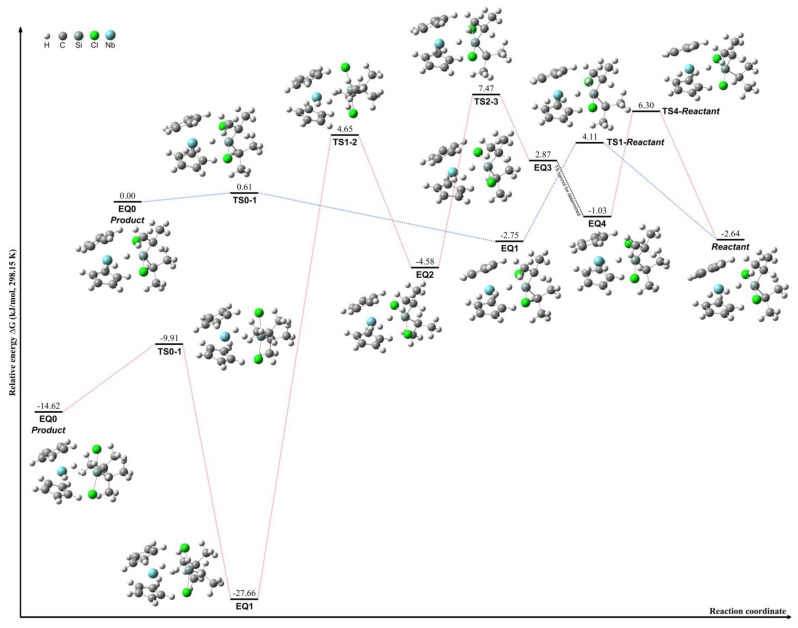
Schematic of the interconversion process between complexes **11** and **12** (red pathway) and hydride migration in complex [Cp_2_NbH_3_]·SiCl_2_*^i^*Pr_2_ (blue pathway).

**Figure 5 molecules-29-05075-f005:**
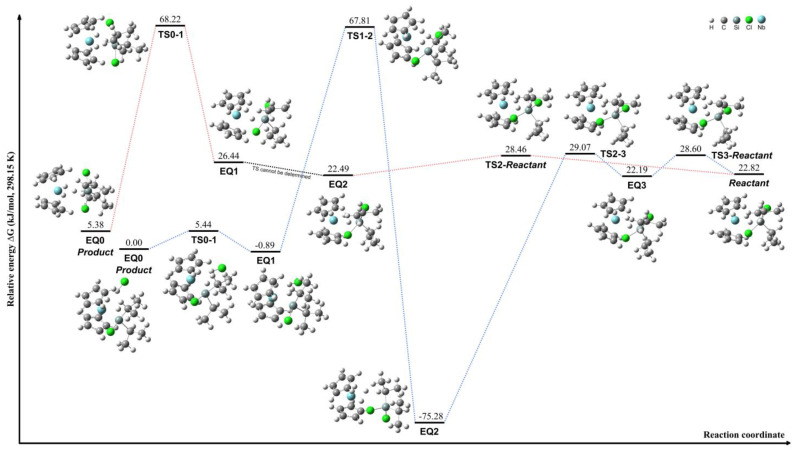
Schematic of the interconversion process between complexes **10** and **11** (red pathway) and the HCl dissociation leading to [Cp_2_NbH_2_]·SiCl*^i^*Pr_2_·HCl intermediates (blue pathway).

**Figure 6 molecules-29-05075-f006:**
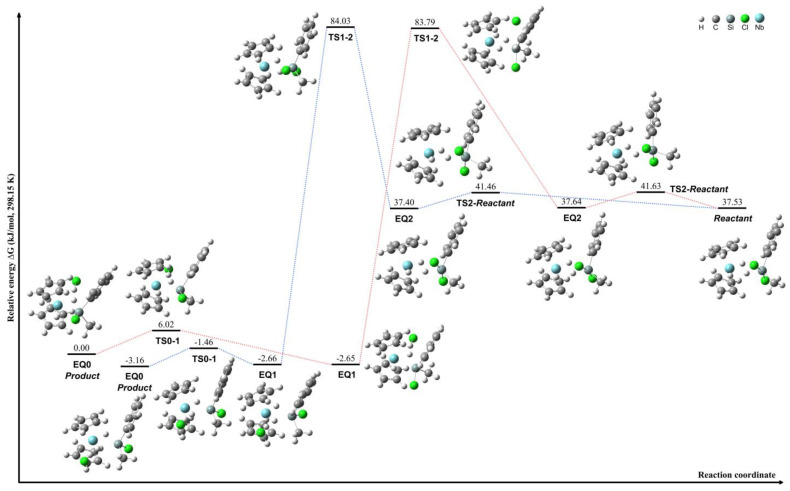
Schematics of the HCl dissociation leading to [Cp_2_NbH_2_·HCl]·SiClMePh intermediates, with pathways illustrated in red and blue.

**Figure 7 molecules-29-05075-f007:**
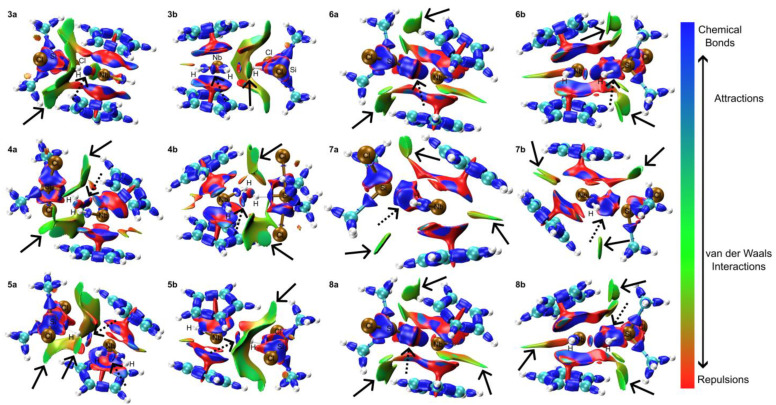
IRI analysis for complexes **1** to **8**. Panels (a and b represent distinct perspectives of the same structure for clarity. Red regions highlight areas of significant repulsion, such as steric effects within rings and cages. Green regions indicate areas dominated by Van der Waals interactions, while blue regions signify notable attractive interactions, including hydrogen bonds, halogen bonds, and chemical bonding. Attractive interactions are shown with dashed arrows, while van der Waals (vdW) forces are represented by solid arrows.

**Figure 8 molecules-29-05075-f008:**
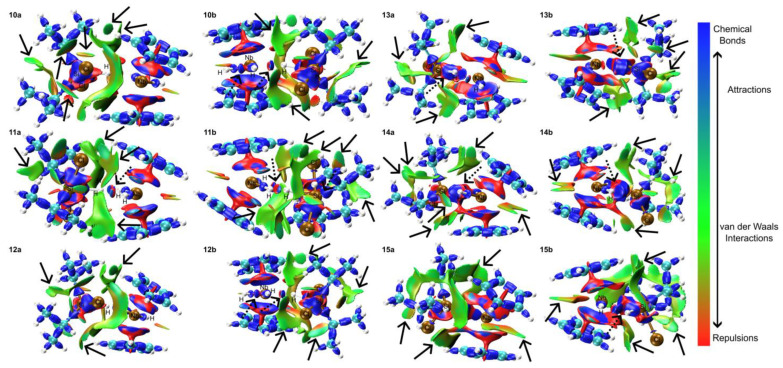
IRI analysis for complexes **10** to **15**. Panels a and b represent distinct perspectives of the same structure for clarity. Red regions highlight areas of significant repulsion, such as steric effects within rings and cages. Green regions indicate areas dominated by Van der Waals interactions, while blue regions signify notable attractive interactions, including hydrogen bonds, halogen bonds, and chemical bonding. Attractive interactions are shown with dashed arrows, while van der Waals (vdW) forces are represented by solid arrows.

**Figure 9 molecules-29-05075-f009:**
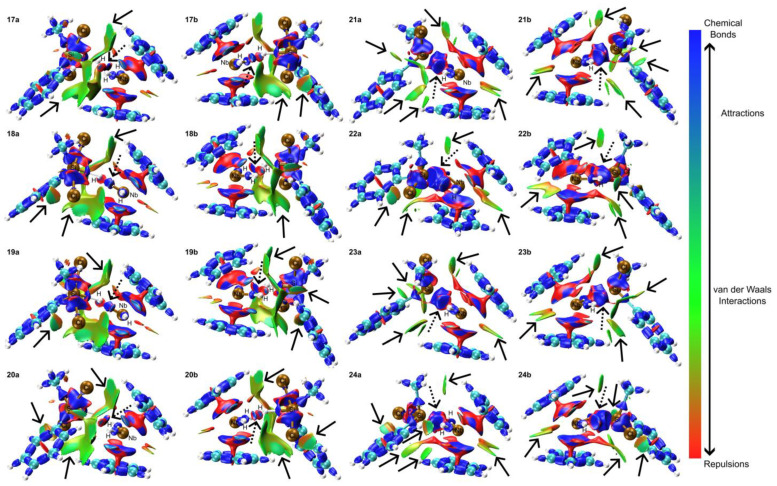
IRI analysis for complexes **17** to **24**. Panels a and b represent distinct perspectives of the same structure for clarity. Red regions highlight areas of significant repulsion, such as steric effects within rings and cages. Green regions indicate areas dominated by Van der Waals interactions, while blue regions signify notable attractive interactions, including hydrogen bonds, halogen bonds, and chemical bonding. Attractive interactions are shown with dashed arrows, while van der Waals (vdW) forces are represented by solid arrows.

## Data Availability

The original contributions presented in the study are included in the article/Appendix A. Further inquiries can be directed to the corresponding author.

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
