# Peer review of "Theoretical Investigation of Interconversion Pathways and Intermediates in Hydride/Silyl Exchange of Niobocene Hydride–Silyl Complexes: A DFT Study Incorporating Conformational Search and Interaction Region Indicator (IRI) Analysis"

_molecules, 2024, doi:10.3390/molecules29215075_

Round 1
Reviewer 1 Report
Comments and Suggestions for Authors
The MS describes theoretical investigation of interconversion pathways and intermediates in hydride/silyl exchange of niobocene hydride-silyl complexes, carried out using various calculation methods. The authors provide a lot of very detailed information obtained on the basis of their calculations. However, after reading the article, it remains unclear what the fundamental problem was and how the authors' work helped to solve this problem and, perhaps, offer recommendations for their implementation in practice.
The main problem should be clearly formulated in the introduction, illustrated by some experimental facts that cannot be simply explained or that contradict something or each other. And only after this does it make sense to present the results of calculations that will help solve the problem. In conclusion, I would like to see not a list of what the authors did (technically), but the authors' achievements in solving those fundamental problems that existed before them and how this can help practicing chemists.
Author Response
Please see the attachement.

Reviewer 2 Report
Comments and Suggestions for Authors
The authors present a comprehensive theoretical study on the subject of complex formation, involving Niobocene hydride and dichlorosilane compounds. In particular, the work is focused on the description of reaction pathways and the possible intertwining of the latter, leading to unexpected results in terms of the yields of the final products. For this purpose, the authors have implemented the artificial force induced reaction (AFIR) method, within the global reaction route mapping GRRM program. Taking into account the variety of theoretical methods and appoaches, used to model reaction mechanisms on molecular level, it would be beneficial if the authors provide a bit of detail about the methods within the GRRM code, which were actively employed in the present research. Going further, it would seem that the GRRM program have several versions developed through the years. The authors should specify which specific version has been used in their research. Finally, a reasoning for the choice of the specific density functional and basis set should be provided in section 2 (Computational Details).
Author Response
Please see the attachement.

Reviewer 3 Report
Comments and Suggestions for Authors
The manuscript was presented in good shape. However, the most stable molecular architectures could be fully describe in order to the chemical bond and the molecular orbital energies with aim to improve the quality of the manuscript and the discussion. In my point of view the intramolecular nature must be revised at the same time that the intermolecular interactions.
Comments on the Quality of English LanguageSome details about the discussion could be revised. However the manuscript is easy to follow.
Author Response
Please see the attachement.

Round 2
Reviewer 1 Report
Comments and Suggestions for Authors
The authors responded to the issues raised by the reviewer and made appropriate changes.
There are only some minor issues:
1) Figure 2 (a): iPr should be instead of Pr.
2) Figure 3: to unify the designations in accordance with how it is given throughout the text, it is better to replace Cl2SiPhCH3 with Cl2SiPhMe.
3) Throughout the text (including in the figures), there is no point in giving energy values with two decimal places; one is enough.
4)
As soon as they correct these the MS can be accepted for publication.
